# Dual Targeting of the EGFR/HER2 Pathway in Combination with Systemic Chemotherapy in Refractory Pancreatic Cancer—The CONKO-008 Phase I Investigation

**DOI:** 10.3390/jcm11164905

**Published:** 2022-08-21

**Authors:** Jana K. Striefler, Jens M. Stieler, Christopher C. M. Neumann, Dominik Geisel, Pirus Ghadjar, Marianne Sinn, Thomas Malinka, Johann Pratschke, Sebastian Stintzing, Helmut Oettle, Hanno Riess, Uwe Pelzer

**Affiliations:** 1Department of Hematology, Oncology and Tumor Immunology, Charité-Universitätsmedizin Berlin, Freie Universität Berlin, Humboldt-Universität zu Berlin, Berlin Institute of Health, 10117 Berlin, Germany; 2II. Medizinische Klinik und Poliklinik, Universitätsklinikum Hamburg-Eppendorf, 20246 Hamburg, Germany; 3Outpatient Department, 16321 Bernau, Germany; 4Department of Diagnostic and Interventional Radiology, Charité-Universitätsmedizin Berlin, Freie Universität Berlin, Humboldt-Universität zu Berlin, Berlin Institute of Health, 10117 Berlin, Germany; 5Department of Radiation Oncology, Charité-Universitätsmedizin Berlin, Freie Universität Berlin, Humboldt-Universität zu Berlin, Berlin Institute of Health, 10117 Berlin, Germany; 6Department of Surgery, Charité-Universitätsmedizin Berlin, Freie Universität Berlin, Humboldt-Universität zu Berlin and Berlin Institute of Health, 10117 Berlin, Germany; 7Outpatient Department, 88045 Friedrichshafen, Germany

**Keywords:** refractory pancreatic cancer, lapatinib, tyrosine kinase, targeted therapy

## Abstract

Background: Primary objective of this present trial was to define the maximum tolerable dose of lapatinib in combination with oxaliplatin, 5-fluorouracil, and folinic acid (OFF) in refractory pancreatic cancer. The secondary objective was to assess the safety and efficacy of lapatinib plus OFF. Methods: We conducted a phase I trial using an accelerated dose escalation design in patients with refractory pancreatic cancer. Lapatinib was given on days 1 to 42 in combination with folinic acid 200 mg/m^2^ day + 5-fluorouracil 2000 mg/m^2^ (24 h) on days 1, 8, 15, and 22, and oxaliplatin 85 mg/m^2^ days 8 and 22 of a 43-day cycle (OFF). Toxicity and efficacy were evaluated. Results: In total, eighteen patients were enrolled: dose level 1 (1000 mg) was assigned to seven patients, dose level 2 (1250 mg), five patients; and dose level 3 (1500 mg), six patients. Dose-limiting toxicities were diarrhea and/or neutropenic enterocolitis observed in two of six patients: one diarrhea III°, one diarrhea IV°, as well as neutropenic enterocolitis. The maximum tolerable dose of lapatinib was 1250 mg OD. Conclusions: The combination of lapatinib 1250 mg OD with platinum-containing chemotherapy is safe and feasible in patients with refractory pancreatic cancer and warrants further investigation.

## 1. Introduction

Pancreatic ductal adenocarcinoma (PDAC) is an aggressive tumor entity expected to become the second leading cause of cancer-related mortality within this decade [1,2]. Treatment of non-resectable patients typically consists of doublet or triplet chemotherapy regimens [3,4,5]. Targeted therapy in pancreatic cancer has been successfully evaluated with an EGFR-inhibitor (erlotinib), but was not implemented as a standard of care, mostly due to the marginal benefit in efficacy and toxicity. A small subgroup of PDAC with BRCA alterations is eligible for PARP inhibition without a clear signal of improved overall survival [6]. 

Currently, there are two standard therapies for patients fit to receive a combination therapy for inoperable PDAC: gemcitabine in combination with nab-paclitaxel or the more intensive FOLFIRINOX regimen, which is reserved for fit patients. In case of progression during first-line treatment with gemcitabine +/− nab-paclitaxel, a subsequent treatment usually comprises a fluoropyrimidine-based combination with either oxaliplatin or (nanoliposomal) irinotecan if an adequate performance status is maintained [7,8,9].

Taken together, a major challenge in the treatment of pancreatic cancer is the absence of effective targeted substances, as well as the lack of validated predictive biomarkers that might facilitate therapeutic decision-making for the vast majority of patients. 

Whereas the addition of the EGFR-targeting agent erlotinib to gemcitabine has been demonstrated to slightly improve outcomes compared with gemcitabine monotherapy [10], the tyrosine kinase inhibitor lapatinib, not only affecting the EGF receptor but also erb-B2, might provide additional benefit. Erb-B2/EGFR heterodimers have a higher tyrosine kinase activity than EGFR homodimers [11]; thus, it might be more efficient to target both receptors. The expression level of the EGFR receptor in PDAC is about 30 to 90 percent, while erbB-2 expression is about 10 to 80 percent in pancreatic cancer tissue samples [12,13,14,15,16,17]. In combination with 5-fluorouracil derivatives, lapatinib might have synergistic effects, as previously shown in a large trial in breast cancer [18,19].

We conducted a phase I trial to define the maximum tolerated dose of lapatinib in combination with platinum containing chemotherapy (OFF regimen) in PDAC patients pretreated with gemcitabine-based therapy.

## 2. Materials and Methods

### 2.1. Study Oversight

This open-label, single-center phase I study investigated lapatinib in combination with OFF in patients with PDAC refractory to gemcitabine monotherapy. The primary objective of the trial was the definition of the maximum tolerable dose of lapatinib in combination with OFF; secondary objectives were safety and efficacy. 

The starting dose of lapatinib was 1000 mg (dose level 1) and subsequently escalated in 250 mg increments. Initiation of part 2 (dose level 2) and part 3 (dose level 3) was performed if none of the three consecutive patients at the respective dose level developed dose-limiting toxicity within the first 42-day cycle of chemotherapy. Observed toxicities were graded according to the National Cancer Institute Common Terminology Criteria for Adverse Events 4.0 (CTCAE). The RECIST (version 1.1) guideline was used for the evaluation of antitumor activity.

The trial protocol was approved by the local ethical committee and registered by the European authorities (EudraCT-Nr. 2009-009928-37). The study was conducted in accordance with the Declaration of Helsinki. All patients provided written informed consent before any study procedures.

### 2.2. Patient Eligibility

Included were patients with histologically proven pancreatic adenocarcinoma and progressive disease during first-line treatment confirmed by CT within the previous 4 weeks. The prerequisite was a normal cardiac function in cardiac ultrasound, normal liver, bone marrow and renal function, a Karnofsky performance status of ≥60 percent, age over 18 years, and written informed consent. The most important exclusion criteria were any history of cardiac arrhythmia, cardiac insufficiency grade NYHA 2 to 4, a history of coronary events, thromboembolic events or cerebral bleeding within the previous 6 months, and prior irradiation. 

Patients were considered noneligible for statistical analysis if a violation of inclusion criteria was present. The remaining patients were included in the intention-to-treat-analysis of primary and secondary objective analysis. According to protocol analysis, only patients receiving ≥2 cycles of chemotherapy were included (exception: dropout due to early progression/death). In terms of toxicity, all patients receiving at least one cycle of chemotherapy and/or targeted therapy were considered evaluable.

### 2.3. Study Design

This phase I trial was designed as a single-institution, prospective, open-label study of the safety and tolerability of lapatinib administered concurrently to the OFF regimen. Enrolment followed the Simon’s accelerated titration design. Three subjects were accrued in the first stage and completed at least one cycle of study treatment. If none of the three consecutive patients at the respective dose level developed dose-limiting toxicity within the first 42-day cycle of chemotherapy, the dose of lapatinib was escalated. In the case of a DLT, three additional patients were included in the respective dose level. If a DLT occurred in ≥2 patients, the prior dose level was considered as the maximum tolerated dose.

Systemic chemotherapy was based on established doses for folinic acid 200 mg/m^2^ (30 min) and 5-FU 2000 mg/m^2^ (24 h continuous infusion) [20]. Both drugs were delivered on days 1, 8, 15, and 22 of a 43-day cycle, whereas oxaliplatin at a dose of 85 mg/m^2^ (2 h infusion) was given on days 8 and 22. Lapatinib was applied daily per os with an initial dose of 1000 mg that was planned to be escalated stepwise by 250 mg to a maximum of 1500 mg. 

To be judged as DLT, a relationship to the study drug was presumed. Therapeutic responses were to be considered confirmed if shown in two subsequent CT scans in a 6-week interval.

### 2.4. Patient Evaluation and Assessments

Response to treatment was evaluated in 6-week intervals by CT scan. For confirmation of response or stable disease, two subsequent routinely performed CT evaluations had to show the respective result. Treatment-related cardiotoxicity was monitored every 6 weeks by cardiac ultrasound and electrocardiogram.

### 2.5. Statistical Analyses

The baseline and treatment characteristics of the eligible patients in the trial who received at least one dose of chemotherapy and/or targeted therapy were summarized. All patients lost to study at any stage following registration (including those not receiving the study medication) were reported. The reason for withdrawal was documented and summarized.

The MTD and rates of DLTs and tolerability were determined at the end of the dose safety and escalation stage of the study. Rates of DLTs are reported with 95% confidence intervals by the dose level cohort in dose safety and escalation stage (based on the dose received). In order to avoid bias due to the occurrence of DLT with subsequently influencing recruitment, the rates of DLT are also reported for the patients in the expansion cohorts.

For the main and final reports after completion of dose escalation, rates of tolerability and grade of adverse events, both reported and only those believed to be possibly, probably, or definitively related to study treatment, were calculated with 95% confidence intervals, overall patients, for those treated at the MTD, and for those treated below the MTD.

## 3. Results

### 3.1. Patient Characteristics

In total, 18 patients allocated to three different dose cohorts were required to determine the maximum tolerable dose (MTD) of the combination regimen. For baseline characteristics and patient assignment criteria, see Table 1A,B and Figure 1, respectively.

#### 3.1.1. Dose Level 1: 1000 mg Lapatinib

Seven patients were included in the lowest dose level. In the therapeutic course of the first three patients, relevant complications occurred. One patient suffered from transitory ischemic neurologic deficit due to carotid arteriosclerosis and interrupted therapy within cycle 1. One patient did not complete the first cycle according to protocol by reason due to hospitalization with hypostatic syncope, diarrhea I°, and hypokalemia III° after vomiting (gastric outlet stenosis); another patient developed fatal liver failure within the first cycle. Initially, it was unknown that the liver failure was a result of severe septicemia (cholangitis and central vein catheter infection) and not possible drug-induced liver toxicity. Thus, we raised the number of patients at the first dose level to acquire sufficient safety data. After receiving all the safety data on the fatal case, it was considered not drug-related but due to sepsis, and further recruitment on this level was stopped. Therefore four instead of three patients were needed to complete this dose level without a DLT. 

#### 3.1.2. Dose Level 2: 1250 mg Lapatinib

In total, five patients received dose level 2. In one of those patients, treatment was stopped within the first cycle when the patient withdrew his consent for individual reasons. Another patient developed progressive disease with hepatic failure within the first cycle. Both patients were considered not to be evaluable for DLTs because they did not complete the first cycle. Thus, two additional patients had to be recruited. We observed no DLT at this dose level. 

#### 3.1.3. Dose Level 3: 1500 mg Lapatinib

Six patients were included in the dose level 3 cohort. The cohort was expanded after the first documented DLT. Overall, two patients experienced a DLT (one III° diarrhea; one IV° diarrhea and neutropenic enterocolitis). For details on the toxicities during the first cycle, please refer to Table 2.

### 3.2. Safety

#### 3.2.1. Toxicity Summary

In order to avoid cumulative toxicities, the three different dose levels were separately given for the subsequent cycles. It has to be considered for interpretation that the number of the given cycles and patients receiving further cycles were necessary. Overall, in dose level 1, 14 subsequent cycles were given (1–8/patients, median 2) to six remaining patients. For dose level 2, six subsequent cycles (1–3/patients, median 2) were given to three remaining patients, and for dose level 3, 12 subsequent cycles (2–4/patients, median 2) were given to four remaining patients. An overview of the given cycles is shown in Figure 1. Table 3 shows the encountered toxicities over the course. Most toxicities were limited to grade II. For details on the more severe toxicities, please refer to the comment on encountered SAEs during the trial.

#### 3.2.2. Toxicity Summary

Thirteen serious adverse events (SAEs) were reported in the course of the trial in nine different patients. Most of the events qualified as SAE through patient hospitalization.

One patient experienced a syncope during cycle 1, resulting in hospital admission due to vomiting, dehydration, hypokalemia, and concomitant QT prolongation that improved after fluid and electrolyte substitution. In the same cycle, the patient was again hospitalized with vomiting, hypokalemia, and mild diarrhea as well as QTc time prolongation, which was shown to be caused by gastric outlet stenosis due to local progression of his carcinoma. The data safety board considered that both events were neither related to the study drug nor protocol procedures. 

One patient suffered from cerebral ischemia due to carotid stenosis within cycle 1 and a traumatic femoral neck shaft fracture during cycle 2, which were both not considered to be related to the study drug or to protocol procedures. 

One patient developed septicemia, most likely due to cholangitis and simultaneous catheter infection with consecutive thrombocytopenia, leukocytopenia, liver failure and respiratory failure—he died during cycle 1. Blood cultures were positive for lactobacillus and candida glabrata. Thus, an abdominal focus, e.g., cholangitis, seemed probable. The fatal event was considered to be related neither to the study drug nor to protocol procedures.

One patient experienced severe hypoglycemia while on insulin therapy in cycles 2 and 4, which were both not considered to be related to the study drug nor protocol procedures.

One patient developed gastrointestinal bleeding III° without concomitant thrombocytopenia but concomitant transaminase elevation IV° that was considered to be caused by hypoxic liver damage during cycle 1. The event was due to cancer infiltration of the stomach and thus not judged to be drug-related or to protocol procedures. 

One patient was hospitalized during cycle 1 for a positive stool test for occult blood; gastroscopy showed no signs of bleeding, and further stool tests were negative. The patient also showed hypokalemia, probably due to pre-existing mild diarrhea. Both events were not considered related to the study drug or to protocol procedures. The same patient was later in the same cycle hospitalized again for cholangitis I°, antibiotics were given and biliary drainage was performed. There was also no relation to the study drug or protocol procedures.

One patient developed diarrhea IV° and neutropenic enterocolitis during cycle 1, which was considered related to lapatinib and possibly 5-FU and oxaliplatin and thus considered to be a DLT. 

One patient was shortly hospitalized in cycle 3 with hyperglycemia and lethargy, most likely due to a new start of parenteral nutrition; both were not considered to be related to the study drug nor protocol procedures. 

One additional patient developed hyperglycemia during cycle 1 after the start of parenteral nutritional support.

## 4. Discussion

In this trial, we found the fixed combination of systemic chemotherapy with oxaliplatin, 5-fluorouracil, and folinic acid combined with a dose of 1250 mg lapatinib daily to be a tolerable combination. Diarrhea III° in one patient and diarrhea IV°, accompanied by febrile enterocolitis in another patient, was defined as a DLT at the dose level of 1500 mg. We consider the dose of 1250 mg as the maximum tolerated dose (MTD) for this combination.

Diarrhea was already the dose-limiting toxicity in the CONKO 003 study establishing the second-line combination with oxaliplatin/5-FU/FS. Doses of lapatinib higher than 1250 mg seem to aggravate this specific side effect. In line with our results, a phase I trial of the combination of lapatinib and capecitabine in patients with solid cancers also observed diarrhea to be the DLT at a combination of lapatinib 1500 mg and capecitabine 2000 mg/m^2^. The MTD was set on lapatinib as 1250 mg daily and capecitabine as 2000 mg/m^2^.

Another trial evaluating lapatinib and FOLFOX4 in patients with solid cancers found no dose-limiting toxicities up to a lapatinib dose of 1500 mg daily, which is above the MTD found in our trial. However, in the cited trial, DLT was defined differently. In the 1500 mg dose level of that trial, 7 of 28 patients experienced diarrhea III°, but this was only considered to be a DLT in case of maximum supportive care [21].

Evaluating the combination of oxaliplatin 130 mg/m^2^ every three weeks together with capecitabine 1500 mg/m^2^ on days 1–14 and lapatinib in diverse solid cancers, Dennie et al. also found diarrhea as the dose-limiting toxicity, and the daily dosage of 1000 mg lapatinib was the maximum tolerated dose [22]. In our trial, the MTD of lapatinib was higher, but there was also a lower dose of oxaliplatin and a continuous 24 h infusion of 5-FU as the combination partner used, which might translate into better gastrointestinal tolerability. A phase 1 trial evaluating lapatinib in combination with either gemcitabine/oxaliplatin or gemcitabine only demonstrated a daily dose of 1500 mg lapatinib in combination with gemcitabine and a daily dose of 1000 mg in combination with gemcitabine/oxaliplatin to be the maximum tolerated dose with nausea and anorexia as DLTs [23]. Interestingly, nausea and anorexia presented no major toxicities in our investigation. 

In conclusion, diarrhea seems to be the most prominent toxicity of lapatinib in combination with fluoropyrimidines with or without oxaliplatin, and, in accordance with most other trials, a daily dose of 1250 mg lapatinib was well-tolerated in our trial.

Cardiac toxicity is a side effect of many tyrosine kinases and a common side effect of the erbB-2 targeting antibody trastuzumab. In the small number of analyzed patients, one patient showed a transient decrease in ejection fraction, possibly related to lapatinib in cycle 3 of dose level 1, which recovered after stopping the study drug. In this patient, study participation was ended due to progressive disease at the same time. Lapatinib is, as trastuzumab, known to provoke mostly transient decreases in ejection fraction [21], although this side effect is not as frequent and pronounced as in trastuzumab [24], and clinical studies have reported the occurrence of symptomatic cardiac impairment in about 0.5% of patients [25].

In the gemcitabine/erlotinib trial, one of the major findings was the prediction of response probability based on rash. The pathophysiology of rash is not yet understood in detail, but delayed maturing of keratinocytes and thinning of superficial skin layers, immune response, and possibly individual differences in response to EGFR inhibition by PI3 kinase activation are discussed [26]. Interestingly, in this trial, no significant rash was found, and usually, rash seems to be less pronounced in erb-B1/erb-B2 inhibitors than in pure erbB-1 inhibitors as erlotinib. A study evaluating these differences in skin specimens found less epidermal atrophy and neutrophilic infiltrations in skin specimens of patients treated with dual inhibitors compared to erbB-1 inhibitors alone as well as an increased expression of pAKT and a decreased dermal expression of the proliferation marker K27 and the negative growth regulator p27 [27].

No higher responses than stabilization were documented in this trial. However, a median OS of 7.6 months after the progression of first line treatment seems to be at least equivalent to the published survival data of the available Phase III second-line trials [7,9].

## 5. Conclusions

The most prominent toxicity of lapatinib in combination with fluoropyrimidines with or without oxaliplatin is gastrointestinal toxicity: diarrhea. Our trial showed that the combination of lapatinib 1250 mg OD with platinum-containing chemotherapy is safe and feasible in patients with refractory pancreatic cancer and warrants further investigation.

## Figures and Tables

**Figure 1 jcm-11-04905-f001:**
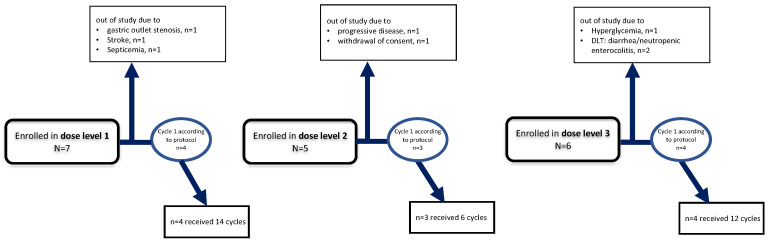
CONKO 008: flow chart.

**Table 1 jcm-11-04905-t001:** (**A**) Baseline patient characteristics. (**B**) Baseline tumor characteristics.

(**A**)	
**Characteristic**	
No. of patients, *n* =	18
Age, median (range), years	62 (50–75)
Karnofsky Performance Status, median (range), %	80 (70–90)
Female, *n*	6
Male, *n*	12
BMI median (range)	18.3 (16.1–27.9)
Previous 1st line therapies	
Gemcitabine	7
Gemcitabine + Erlotinib	6
Gemcitabine + Aflibercept	13
Gemcitabine + Sorafenib	3
Gemcitabine + Capecitabine	1
(**B**)	
**Tumor Characteristic**	
Stage	
*localized n*	0
*metastasized n*	18
Initially curative intended resection	
yes, *n*	9
*no*, *n*	9
Histology	
*Adenocarcinoma ductal*	17
*Adenocarcinoma papillary*	1
Tumor Grading	
*G1*	0
*G2*	12
*G3*	6
*Pattern of Progression*	
pulmonary	5
liver	7
both	9

**Table 2 jcm-11-04905-t002:** Toxicities in cycle 1, number of maximum toxicities per cycle and patient.

	Dose Level 1 (Lapatinib 1000 mg)				Dose Level 2 (Lapatinib 1250 mg)				Dose Level 3 (Lapatinib 1500 mg)			
CTC AE 4.0 Grade	I	II	III	IV	I	II	III	IV	I	II	III	IV
Anemia	6	1	0	0	3	1	1	0	4	1	0	0
Leukopenia	2	0	0	0	1	0	0	0	0	0	0	0
Neutropenia	1	0	1	0	1	0	0	0	0	0	0	0
Febrile neutropenia	0	0	0	0	0	0	0	0	0	0	1	0
Thrombocytopenia	2	0	0	1	1	0	0	0	0	0	0	0
Potassium	2	1	1	0	0	0	2	0	1	0	1	0
Sodium	2	0	0	0	2	0	1	0	1	0	1	0
Albumin	3	0	0	0	1	3	0	0	1	3	0	0
ALT	1	0	1	0	2	0	1	0	1	0	0	0
AST	1	0	0	1	3	0	0	1	1	0	0	0
GGT	2	2	2	0	2	1	2	0	0	1	2	1
Alkaline phosphatase	0	2	0	0	4	0	0	0	0	0	1	0
Creatinine	1	0	0	0	1	0	0	0	0	0	0	0
Bilirubin	2	0	0	1	1	0	0	0	0	0	1	0
QTc-time	1	0	0	0	0	0	0	0	0	0	0	0
Hypertension	0	3	0	0	1	2	0	0	3	2	1	0
Nausea	3	2	0	0	2	1	0	0	3	1	0	0
Vomiting	2	1	0	0	1	0	0	0	1	0	0	0
Diarrhea	4	1	0	0	2	2	0	0	1	2	1	1
Infection	0	0	0	1	1	0	0	0	0	0	0	0
Fatigue	3	2	0	0	3	2	0	0	1	2	0	0
Hand-Foot Syndrome	1	0	0	0	1	0	0	0	3	0	0	0
Synkope	0	0	1	0	0	0	0	0	0	0	0	0
Gastric outlet stenosis	0	0	1	0	0	0	0	0	0	0	0	0
Stroke	0	0	1	0	0	0	0	0	0	0	0	0
GI bleeding	0	0	1	0	0	0	0	0	0	0	0	0
Hyperglycemia	0	0	0	0	0	0	0	0	0	0	0	1
Pain	1	2	1	0	3	0	2	0	1	0	0	0

Heat map: The color intensity indicates the number of patients with toxicity. The darker the color, the higher the number of patients.

**Table 3 jcm-11-04905-t003:** Toxicities in subsequent cycles, based on maximum toxicity per patient and cycle.

Toxicity	Dose Level 1: Lapatinib 1000 mg				Dose Level 2: Lapatinib 1250 mg				Dose Level 3: Lapatinib 1500 mg			
CTC AE 4.0 Grade	I	II	III	IV	I	II	III	IV	I	II	III	IV
Anemia	8	7	0	0	4	2	0	0	10	0	0	0
Leukopenia	6	0	0	0	2	1	0	0	3	0	0	0
Neutropenia	3	0	0	0	0	0	2	0	1	0	0	0
Febrile Neutropenia	0	0	0	0	0	0	0	0	0	0	0	0
Thrombocytopenia	5	1	0	0	3	1	0	0	1	0	0	0
Potassium	4	1	2	0	1	0	1	0	2	0	0	0
Sodium	3	0	0	0	1	0	2	0	4	0	0	0
Calcium	5	7	0	0	2	3	1	0	5	1	0	0
Magnesium	9	0	0	0	6	0	0	0	4	0	0	0
Albumin	5	4	1	0	2	4	0	0	5	0	0	0
ALAT	3	1	0	0	1	1	0	0	3	0	0	0
ASAT	7	0	0	0	4	0	0	0	3	0	0	0
GGT	9	2	2	0	3	2	1	0	1	1	2	0
Alkaline phosphatase	8	0	0	0	4	0	0	0	3	1	1	0
Creatinine	0	0	0	0	0	0	0	0	0	0	0	0
Bilirubin	0	0	0	0	0	0	0	0	0	0	0	0
Ejection fraction	0	1	0	0	0	0	0	0	0	0	0	0
QTc-time	1	0	0	0	0	1	0	0	0	0	0	0
Hypertension	3	4	0	0	3	2	1	0	5	1	0	0
Nausea	4	2	0	0	1	0	0	0	5	2	0	0
Vomiting	4	1	0	0	0	0	0	0	2	0	0	0
Diarrhea	6	3	2	0	3	1	0	0	3	2	1	0
Infection	0	0	0	0	0	0	0	0	0	0	0	0
Fatigue	4	7	0	0	3	3	0	0	6	5	0	0
Hand-Foot-Syndrome	7	1	0	0	2	0	0	0	7	0	0	0
Hyperglycemia	0	0	0	0	0	0	0	0	0	0	1	0
Pain	2	1	1	0	2	1	2	0	6	0	0	0
Fracture	0	0	1	0	0	0	0	0	0	0	0	0
Hypoglycemia	0	0	0	0	0	0	2	0	0	0	0	0
Thromboembolism	0	0	0	0	0	0	1	0	0	0	0	0

Heat map: The color intensity indicates the number of patients with toxicity. The darker the color, the higher the number of patients.

## Data Availability

The datasets generated during and/or analyzed during the current study are available from the corresponding author on reasonable request.

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
