# Peer review of "Dual Targeting of the EGFR/HER2 Pathway in Combination with Systemic Chemotherapy in Refractory Pancreatic Cancer—The CONKO-008 Phase I Investigation"

_jcm, 2022, doi:10.3390/jcm11164905_

Round 1
Reviewer 1 Report
Authors present an interesting phase 1 study to determine the toxicity of the combination of lapatinib in combination with FOLFOX in increasing doses. After the inclusion of 18 patients, of whom 4 received dose 1 effectively, three received dose two and 6 apparently received dose three. The work is interesting and can lay the foundations of the appropriate dose for later clinical trials. However, there are some issues that are not well clarified.
1. The authors conclude that diarrhea is the main toxic effect of the drug combination. What is this statement based on? On the one hand, it is not the most frequent effect in any of the three dose levels, nor is it the most serious. Of the 13 events considered serious, only one was diarrhea caused by the combination of drugs. Only diarrhea was considered DLT? The rest of the adverse effects were not considered DLT? What was this distinction based on?
2. Secondary objectives of the study were safety and efficacy. How did you meassure efficacy ? Were are the resiults reported?
3. Although a case of grade IV toxicity appeared at dose level 3, there were also grade 4 complications for dose level three, and many more grade 3. The statement, with these results, that the dose level 2 is the maximum tolerable?
Author Response
Dear colleague,
thank you very much for your interest and critical questions, i will try to answer and clarify them below:
1)
We also documented diarrhea as a DLT in the evaluation of the CONKO 003 study which explored the combination of oxaliplatin and 5-FU. We have therefore set preferences here, furthermore we focused the side effects on clinical and quality-of-life-limiting symptoms.
At dose level 3, we documented grade III and grade IV toxicity in the initial cycle, necessitating hospitalization (Table 2). This is therefore to be determined as a DLT. This toxicity no longer occurred in subsequent cycles after the dose had been adjusted (Table 3).
2)
We have determined the effectiveness as a secondary goal as a classic method using imaging methods (line 116). These were carried out every 6 weeks and evaluated and discussed in our cancer conference. The results were mentioned in the discussion (line 297)
3)
I assume that you mean that higher toxicities also occurred at dose level II, which were not assessed as DLT. That's right, there was a safety monitoring committee that clarified the assessment of toxicity in terms of context. If there was disease-related and not therapy-related toxicity, that tox was not assessed as a DLT. We have tried to present the individual toxicities in detail in section 3.2.
Reviewer 2 Report
The manuscript entitled 'Dual targeting of the EGFR/HER2 pathway in combination with systemic chemotherapy in refractory pancreatic cancer -The CONKKO-008 Phase 1 Investigation' by Striefler and coauthors deals with the Phase I trial examination. The study's primary objective is to assess the safety and efficacy of chemotherapy using an accelerated dose escalation design in patients with refractory pancreatic cancer. Phase I trials are a critical first step of therapeutic attempts. The primary goals are to find recommended dose, schedule, and pharmacological behavior of new agents or new combinations and determine the adverse effects of treatment. Hence, this study covers a few aspects and provides the combination of lapatinib with platinum is safe and feasible for patients. The present research is promising and appealing.
Abstract looks okay, but sentences are fragmented, complicated, and associated with grammatical errors that need to be corrected. Likely, (line no. 21-23) Primary objective…… needs to be correct as the primary objective of this present trial was to define the maximum ……Secondary objective was to assessment…….
In addition, the authors need to remove numbering in the Abstract section.
The introduction section could be elaborated on some points, such as explaining the therapeutic failure of a single targeting effect of the EGFR/HER2 pathway and its rationale with other clinical trial observations. Line 44 -needs to cite the reference.
The results and discussion sections are well interpreted and explained, but lots of sentence fragmentation with grammatical errors. Discussion section – need to correct typo errors such as line no. 285 about 0,5% ? of patients…..
Line no. 299. typo error Phase III second-line trials [6], [8].5.?
Figure 1 is not adequately uploaded means the half section is missing for defining the Patient's assignment criteria.
Tables 2 and 3 contain the orange color grading that must be clearly defined in table legends.
The table 1B, the authors can show the grading in CTCAE 4.0 as tables 2 and 3 data represented similarly, it would provide ease for the readers.
In study design section (line no. 109) authors mentioned the 43 day cycle, while in Abstract (line no 25) denoted 42 day cycle. Please make it clear.
In results section 3.1.1. and 3.1.2 the Patient who showed liver or hepatic toxicity were excluded from the study. The provided explanation by the authors is due to septicemia, not drug-induced toxicity. On which basis or lab testing are authors suggesting this? The authors did not mention the liver /hepatic history of the patients.
Do authors analyze the cardiotoxicity of the drug? It needs to mention the toxicity doses if the past study has already been reported.
Author Response
Thank you very much for your interest in our work and for your critical, helpful comments, which we were happy to address.
We have formulated the abstract in a more understandable way according to your suggestions. The numbering has been removed accordingly.
We would have liked to have written more about the suspected escape mechanisms, but we deliberately wanted to keep the introduction short, clear and concise in order to lead directly to the methodology and results. In the discussion, we tried to take up possible effects of a sole unidirectional EFGR/Her2 inhibition. For deeper discussions, we would like to refer to various review articles to avoid redundant information.
Right - the citation of the Polo study was added.
The result and the discussion part were checked again for fragmentation and grammatical errors.
We consider the information on cardiac toxicity to be correct at 0.5% (line no. 284)
Citation: Perez and colleagues reviewed the cardiac safety data in 4990 patients enrolled in 44 clinical trials using lapatinib. In these trials, 3689 (74%) received lapatinib while 1301 patients (26%) served as contemporary controls. LVEF was prospectively evaluated via multiple-gated acquisition (MUGA) scan or echocardiography at screening, every 8 weeks during therapy, and at withdrawal. Cardiac events were defined as symptomatic (grade 3 or 4) heart failure or non symptomatic (LVEF decreases > 20% relative to baseline and below the institution’s lower limit of normal). Of the 3689 study patients, 60 patients (1.6%) experienced a cardiac event, of whom only 7 (0.2%) were symptomatic. Importantly the incidence of cardiac events in the lapatinib-treated patients was more or less similar to that reported in the 1301 patients who did not receive lapatinib (0.7%). Prior treatment with anthracyclines (598 cases) was associated with cardiac dysfunction in 2.2%, which was symptomatic in only 0.3% of the treated patients. Similar result was also seen in patients previously exposed to trastuzumab (1.7%).
Figure 1 was missed by formatting error – correction was done.
Right, we assumed that the coloring of the heat map is self-explanatory, to avoid misunderstandings, we have added a definition. “HEAT MAP: the color intensity indicates the number of patients with toxicity. The darker the color, the higher the number of patients”
You mentioned that: „The table 1B, the authors can show the grading in CTCAE 4.0 as tables 2 and 3 data represented similarly, it would provide ease for the readers.° - Unfortunately, I cannot interpret the criticism, CTC AE interprets side effects of therapies. Classifications for tumor grading are not defined here.
Right – we defined the 43d cycle, d43 cycle 1 = d1 of cycle 2, correction was made.
Im Ergebnisabschnitt 3.1.1. und 3.1.2 der Patient, der eine Leber- oder Lebertoxizität zeigte, wurde von der Studie ausgeschlossen. Die von den Autoren gelieferte Erklärung beruht auf Septikämie, nicht auf arzneimittelinduzierter Toxizität. Auf welcher Grundlage oder Labortests schlagen die Autoren dies vor? Die Autoren erwähnten die Leber-/hepatische Vorgeschichte der Patienten nicht.
The described patient in 3.1.1 with fatal liver failure was initially considered to be a DLT for safety reasons, so the cohort was expanded. After the fatal death, the patient was autopsied; a venous port infection and cholangitis were detected as well as typical septic changes and embolization’s/necrosis in the organs, primarily in the liver. The data monitoring board has thus determined the causal connection to the sepsis and not primarily due to the combination of drugs.
As already shown in tables 2 and 3, the cardiac function was monitored as recommended in label, the QTC and ejection fraction were measured and the toxicities were presented. Highest toxicities were at grade 2 and showed no dose relation.